# Multi-Omic Analysis of CIC’s Functional Networks Reveals Novel Interaction Partners and a Potential Role in Mitotic Fidelity

**DOI:** 10.3390/cancers15102805

**Published:** 2023-05-17

**Authors:** Yuka Takemon, Véronique G. LeBlanc, Jungeun Song, Susanna Y. Chan, Stephen Dongsoo Lee, Diane L. Trinh, Shiekh Tanveer Ahmad, William R. Brothers, Richard D. Corbett, Alessia Gagliardi, Annie Moradian, J. Gregory Cairncross, Stephen Yip, Samuel A. J. R. Aparicio, Jennifer A. Chan, Christopher S. Hughes, Gregg B. Morin, Sharon M. Gorski, Suganthi Chittaranjan, Marco A. Marra

**Affiliations:** 1Genome Science and Technology Graduate Program, University of British Columbia, Vancouver, BC V5Z 4S6, Canada; ytakemon@bcgsc.ca; 2Canada’s Michael Smith Genome Sciences Centre, BC Cancer Research Institute, Vancouver, BC V5Z 1L3, Canada; vleblanc@bcgsc.ca (V.G.L.); annie.moradian@cshs.org (A.M.); sgorski@bcgsc.ca (S.M.G.);; 3Department of Pathology & Laboratory Medicine, University of Calgary, Calgary, AB T2N 1N4, Canada; 4Arnie Charbonneau Cancer Institute, University of Calgary, Calgary, AB T2N 4Z6, Canada; 5Alberta Children’s Hospital Research Institute, University of Calgary, Calgary, AB T2N 4N1, Canada; 6Department of Clinical Neurosciences, University of Calgary, Calgary, AB T2N 1N4, Canada; 7Department of Molecular Oncology, BC Cancer Research Institute, Vancouver, BC V5Z 1L3, Canada; syip-02@bccancer.bc.ca (S.Y.); saparicio@bccrc.ca (S.A.J.R.A.); chughes@bccrc.ca (C.S.H.); 8Department of Pathology and Laboratory Medicine, University of British Columbia, Vancouver, BC V6T 1Z7, Canada; 9Department of Medical Genetics, University of British Columbia, Vancouver, BC V6H 3N1, Canada; 10Department of Molecular Biology and Biochemistry, Simon Fraser University, Burnaby, BC V5A 1S6, Canada

**Keywords:** CIC, genetic networks, proteomic interactions, single-cell sequencing, cell cycle, mitosis, splicing

## Abstract

**Simple Summary:**

Capicua (*CIC*) is a gene that is frequently mutated in several cancer types, including stomach cancers and certain subtypes of brain tumours and sarcomas. CIC, the protein encoded by the *CIC* gene, has been shown to play a multitude of roles in both normal and cancer cell functions; however, most studies exploring these roles focus on a single aspect of CIC function and may therefore overlook complex interconnected activities in which CIC is involved. In this study, we have used multiple complementary approaches to obtain a broader view of CIC’s complex functional networks. We observed novel interactions (genetic or physical) between CIC and genes/proteins involved in various aspects of cellular function, including regulation of cell division and processing of RNA molecules. Altogether, our work characterises the complexity of CIC’s functional network and expands our understanding of its potential roles in cancer.

**Abstract:**

*CIC* encodes a transcriptional repressor and MAPK signalling effector that is inactivated by loss-of-function mutations in several cancer types, consistent with a role as a tumour suppressor. Here, we used bioinformatic, genomic, and proteomic approaches to investigate CIC’s interaction networks. We observed both previously identified and novel candidate interactions between CIC and SWI/SNF complex members, as well as novel interactions between CIC and cell cycle regulators and RNA processing factors. We found that CIC loss is associated with an increased frequency of mitotic defects in human cell lines and an in vivo mouse model and with dysregulated expression of mitotic regulators. We also observed aberrant splicing in CIC-deficient cell lines, predominantly at 3′ and 5′ untranslated regions of genes, including genes involved in MAPK signalling, DNA repair, and cell cycle regulation. Our study thus characterises the complexity of CIC’s functional network and describes the effect of its loss on cell cycle regulation, mitotic integrity, and transcriptional splicing, thereby expanding our understanding of CIC’s potential roles in cancer. In addition, our work exemplifies how multi-omic, network-based analyses can be used to uncover novel insights into the interconnected functions of pleiotropic genes/proteins across cellular contexts.

## 1. Introduction

Capicua (CIC) is a transcriptional repressor that is impacted by somatic mutations or rearrangements in several cancer types, including undifferentiated small round cell sarcomas (~60% of cases) [1], oligodendrogliomas (~50–80%) [2,3], gastric adenocarcinomas (~9%) [4], endometrial carcinomas (~7%) [5], and melanomas (~8%) [6,7]. CIC loss has also been linked to prostate cancer [8], lung cancer [9], and T-cell lymphoblastic leukemia/lymphoma [10,11]. *CIC* alterations and/or loss of CIC activity have been associated with inferior survival outcomes in several cancer types [12,13,14,15] and with aggressive phenotypes, such as increased metastatic potential [9] and resistance to kinase inhibitors [16,17,18]. Together, these observations are compatible with the notion that CIC functions as a tumour suppressor in several cancer types.

CIC is a transducer of receptor tyrosine kinase (RTK) signalling and functions through a default repression mechanism in which RTK activation leads to reduced CIC activity and subsequent de-repression of its target genes [19]. Targets of CIC are also themselves enriched for genes involved in the mitogen-activated protein kinase (MAPK) signalling cascade, including effectors (e.g., *PTPN9*, *SHC3/4*) and members of regulatory feedback mechanisms (e.g., *DUSP4/5/6*, *SPRY2/4*, *SPRED1/2/3*) [13,20,21,22,23,24]. The cancer-promoting properties observed in cells with disrupted CIC activity have been explained, at least in part, by the de-repression of one or more of its known target genes that are effectors of MAPK signalling, especially the ETS transcription factors *ETV1*, *ETV4*, and *ETV5* [8,9,10]. In *Drosophila*, where Cic function was originally described [25], Cic inhibits the expression of cell cycle genes (e.g., String/Cdc25, Cyclin E) and is a key mediator of Ras-driven proliferation [26,27]. Although the relationship between RAS/MAPK signalling, CIC function, and proliferation appears more complex in mammalian cells [19], loss of CIC has also been associated with proliferation in the absence of RTK signals in cancer cells and neural stem cells [16,17,22,28] and with dysregulated expression of cell cycle-related genes (e.g., *AURKA*, *CDK4*, *BUB1*) across multiple cancer types [29]. CIC has also been implicated in the development and physiology of multiple mammalian organ systems, including brain and lung development, liver homeostasis, and T-cell differentiation [30]. It has been shown that CIC can interact with proteins such as ATXN1 [31], ATXN1L [29], and SIN3A [22,32] to mediate the expression of its targets. Interestingly, a recent study by Hwang et al. [32] demonstrated that CIC interacts with chromatin remodelers, such as several members of the switch/sucrose non-fermenting (SWI/SNF) complex—namely ARID1A, ARID1B, SMARCC1, SMARCC2, SMARCA2, and SMARCA4—to regulate neuronal differentiation. These observations give rise to the notion that CIC functions are diverse and are mediated through an increasing number of interactors that are only partially identified.

The increasing frequency of CIC-related publications in recent years appears to indicate a heightened level of interest in this apparently pleiotropic protein that, to date, is understood to be involved in transcriptional regulation and signal transduction to mediate phenotypes across a wide array of developmental and oncogenic contexts. Many of these CIC-related studies focus on a singular aspect of its function (e.g., transcriptional repression) in a unique context (e.g., neuronal progenitor cells). The rapid development of genomic techniques and technologies in recent years (e.g., genome-wide CRISPR screens, whole-proteome quantification, single-cell sequencing, etc.), along with access to large public datasets generated by them, have enabled powerful investigations of complex biological relationships from different angles through layering of multiple -omic technologies (e.g., VanderSluis et al. [33], Takemon et al. [34], Shan et al. [35]). To obtain a broader view of CIC function and reveal interactions and relationships that may have remained obscured in more focused analyses, here we make use of large publicly available datasets and generate additional multi-omics data to map CIC’s genetic, proteomic, and transcriptional regulatory networks in human cell lines. We identify several SWI/SNF complex members as genetic and proteomic interaction partners of CIC, reproducing previously identified protein interactions and identifying novel interactions with additional members of the complex. We also identify several cell cycle regulators in CIC’s genetic and proteomic networks and, using single-cell RNA sequencing (scRNA-seq), show that several cell cycle regulators exhibit dysregulated gene expression in CIC-deficient cell lines. Furthermore, using human cell lines and a forebrain-specific *Cic* knockout (KO) mouse model, we demonstrate that loss of CIC is associated with an increased frequency of mitotic defects. Finally, motivated by the observation that CIC’s genetic and proteomic networks include several RNA processing and splicing factors, we perform a differential splicing analysis in which we compare *CIC* wild type (WT) and KO human cell lines and find differential exon usage patterns predominantly at 3′ untranslated regions (UTRs). Overall, our systematic characterization of CIC’s complex functional network reveals the effects of its loss on cell cycle regulation, mitotic integrity, and transcriptional splicing stability, thus providing new insights into CIC’s potential role(s) in different biological contexts, including cancer. We argue that, in addition to revealing novel CIC functions, our study provides a blueprint for multi-omic characterization of pleiotropic genes to uncover novel connections and functions at play across cellular and subcellular contexts.

## 2. Methods

### 2.1. Cell Culture

The IDH1-stable NHA cell line was obtained from Applied Biological Materials (ABM) Inc. (T3022; Richmond, BC, Canada). The HEK293A cell line was obtained from Dr. Gregg Morin (Canada’s Michael Smith Genome Sciences Centre, Vancouver, BC, Canada) and authenticated by Genetica DNA Laboratories (Cincinnati, OH, USA). The HOG cell line was obtained from Dr. G. Dawson (University of Chicago, Chicago, IL, USA). Cells were cultured in Dulbecco’s Modified Eagle Medium (DMEM) supplemented with 10% (*v*/*v*) heat-inactivated (HI) FBS (Thermo Fisher Scientific, Waltham, MA, USA). Stable cell lines expressing F-CIC-S, F-CIC-L, or F-vec were cultured in DMEM + 10% HI FBS with Zeocin (Thermo Fisher Scientific) selection. Cell culture was performed in a humidified, 37 °C, 5% CO_2_ incubator.

### 2.2. CIC Knockout Cell Line Generation and Expression Constructs

CRISPR-Cas9 sgRNA sequences were designed to target exon 2 of the *CIC* gene, and *CIC*^KO^ lines were established from NHA (NHA-*CIC*^KO-A2^ and NHA-*CIC*^KO-H9^) and HEK293A (HEK-*CIC*^KO-A9^ and HEK-*CIC*^KO-D10^) cells as described previously [21]. *CIC* knockout status was verified by Western blot and Sanger sequencing. N-terminal FLAG-CIC-short form (FLAG-CIC-S) and long-form (FLAG-CIC-L) plasmids were prepared as described previously [36].

### 2.3. scRNA-Seq

Single-cell 3′ RNA-seq libraries were generated using the Chromium Single Cell 3′ Library & Gel Bead Kit v1 (10x Genomics, Pleasanton, CA, USA) following the manufacturer’s protocol. Full details can be found in the Appendix A.

The CellRanger software (v1.2.1 and 2.1.1) was used to create fastq files (cellranger mkfastq with default parameters; v1.2.1) and to perform alignment, filtering, barcode counting, and unique molecular identifier (UMI) counting (cellranger count with default parameters except --expect-cells set to 1000 for all samples; v2.1.1). The CellRanger hg19-3.0.0 reference was used for alignment. Data preprocessing was performed in the R statistical software (v3.5.0) [37], based on the filtered count matrices output by CellRanger. Counts for cells from all samples (7735 cells) were combined into a single count matrix, and genes detected (UMI > 0) in at least two cells were retained. Normalisation was then applied to all cells using the scran R package [38] (v1.10.1). The *quickCluster* function was used to cluster cells for normalisation (with min.mean = 0.1 as suggested for UMI data), and the resulting clusters were used as input to the *computeSumFactors* function (with min.mean = 0.1). These factors were then used in the *normalise* function of the scater R package (v1.10.0).

Cells were scored for cell cycle phases as described by Macosko et al. [39] and Schwabe et al. [40] using the phase-specific genes identified by Whitfield et al. [41] (Appendix A). Scores were obtained using the *estimate_Schwabe_stage* function implemented in the tricycle R package [42] (v1.2.1) with default parameters. Only cells that passed the default filtering criteria were used for the differential expression analysis: G1/S, 725 NHA-*CIC*^WT^ cells, 554 NHA-*CIC*^KO-A2^ cells, and 532 NHA-*CIC*^KO-H9^ cells; S, 478, 186, and 270; G2, 237, 66, and 94; G2/M, 200, 226, and 234; M/G1, 660, 111, and 155. For the differential expression analysis, differential expression was calculated using the DESeq2 R package [43] with default parameters comparing all NHA-*CIC*^WT^ vs. NHA-*CIC*^KO^ samples. Genes with a Benjamini Hochberg (BH)-corrected *p*-value < 0.05 and an absolute fold-change > 1.5 were considered to be significantly differentially expressed. To visualise these expression differences, DESeq2-normalised expression values were obtained using the *counts* function with normalised = TRUE. By default, DESeq2 performs independent filtering based on mean expression across samples in order to select a set of genes for multiple test correction that maximises the number of significantly differentially expressed genes found at the specified false discovery rate [43]. Therefore, for each differential expression analysis, genes that were detected below the mean cutoff for that comparison were not analysed (returning a corrected *p*-value of “NA”); these comparisons are indicated with “NC” (not calculated) in the relevant figures.

### 2.4. IP-MS

The IP-MS analyses of nuclear fractions from HEK293A cells using anti-endogenous CIC or anti-Myc (for N-Myc-CIC-S) antibodies (Appendix A) were performed as described previously [36], except using nuclear cell fractions (see Appendix A for more detail). Three replicates were performed against endogenous CIC, and one experiment was performed against N-terminal MYC-fused CIC stably expressed in HEK293A cells. The IP-MS validation experiments performed in nuclear fractions of the NHA cell line were performed in triplicate, as were the matched IgG controls. See Appendix A for methodological details.

### 2.5. Co-Essentiality Mapping and In Silico Genetic Screening Framework

Co-essentiality mapping and in silico genetic interaction screening were performed using the R statistical software (v4.0.2) package GRETTA (v0.5.0) [44], with DepMap public release version 20Q1 [45] data from the DepMap platform portal (https://depmap.org/portal/download/; accessed on 10 February 2020). Briefly, co-essentiality mapping uses fitness effect scores that were derived from normalised and copy number-corrected whole-genome CRISPR-Cas9 knockout screens targeting 18,333 genes performed in 739 cancer cell lines [46,47]. GRETTA calculated and ranked Pearson correlation coefficients between fitness scores of *CIC* and each screened gene, and a gene pair was considered co-essential if its BH-adjusted *p*-value was <0.05 and its coefficient was >0.178 (the inflection point). Given the BH adjustments, we expect a <5% false discovery rate in our analysis. For non-syntenic co-essential maps, genes on chromosome 19 were removed and the thresholds were recalculated (with an inflection point coefficient of 0.138).

Using GRETTA, *CIC*-mutant lines were identified using genomic data from the 739 DepMap cancer cell lines. We identified two lines (HCC366 and JHUEM1) with homozygous deleterious *CIC* alterations (*CIC*^HomDel^), five lines (REH, CCK81, HEC6, GP4D, and SISO) with potentially trans-heterozygous *CIC* mutations (*CIC*^T-HetDel^; multiple heterozygous deleterious *CIC* mutations or a combination of partial copy number loss with heterozygous deleterious *CIC* mutation[s]), 53 lines with a single heterozygous deleterious *CIC* mutation or partial copy number loss (*CIC*^HetDel^), and 557 lines with no *CIC* alterations (*CIC*^WT^; Appendix A). The normalised TPM gene and protein expression values for the selected cell lines were extracted using GRETTA.

Pairwise Mann-Whitney U tests between *CIC* mutant and *CIC*^WT^ cell lines for all 18,333 genes were performed using GRETTA to obtain *p*-values. These *p*-values were adjusted for multiple testing using a permutation approach, whereby the lethality probabilities were randomised and resampled 10,000 times. Candidate genetic interactors of *CIC* that were identified using *CIC*^HomDel^ mutant cell lines were defined as those with an adjusted Mann-Whitney U test *p*-value < 0.05 and a median lethality probability > 0.5 in at least one group. Known essential and non-essential genes, as defined by DepMap, were filtered out.

### 2.6. Co-Expression Analysis

Genes co-expressed with *CIC* were downloaded from COXPRESdb (v8.1) [48]. COXPRESdb is a database that was previously used by Pan et al. [49] and Wainberg et al. [50] to benchmark co-essential genes. The database contains co-expression z-scores between *CIC* and 17,067 genes from over 10,485 human samples assayed using RNA-seq (“hsa-r.6”), where a z-score of zero is a random level and a score of three is a false positive rate of 0.1% for any species or version. In this study, we used the top 50 genes co-expressed with CIC.

### 2.7. ChIP-Seq Analysis

CIC ChIP-seq peaks for NHA cells were obtained from Lee et al. [24] (Appendix A in that publication). ChIP-seq data from mouse embryonic stem cells (mESCs; fastq files) were downloaded from ArrayExpress (ERR2534130 [CIC ChIP] and ERR2534131 [IgG control], data from Weissmann et al. [22]) and SRA (SRR7643337 [ARID1A ChIP replicate 1], SRR7643339 [ARID1A input control replicate 1], SRR7643338 [ARID1A ChIP replicate 2], SRR7643340 [ARID1A input control replicate 2], SRR7643346 [BRD4 ChIP], SRR7643347 [BRD4 input control], SRR6792731 [BRD9 ChIP], SRR6792735 [BRD9 input control], SRR7724001 [SMARCA4 ChIP], and SRR7724002 [SMARCA4 input control]; data from Gatchalian et al. [51]). Alignments were performed using BWA (v0.7.17) [52], using the mm10 genome as a reference. Aligned read files were sorted and indexed using Samtools (v1.9) [53,54]. Peaks were called using MACS2 (v2.1.1) [55], using the input files as controls and default parameters. For ARID1A, peaks that were identified in both replicates (≥1 base overlap) were retained for downstream analyses. Peaks were annotated using the annotatePeak function from the ChIPseeker R package (v1.30.3) [56,57] using the TxDb.Mmusculus.UCSC.mm10.knownGene annotation from the R package of the same name (v3.10.0). The findOverlaps function from the GenomicRanges R package (v1.46.1) [58] was used to identify CIC peaks that overlapped with one or more SWI/SNF peaks. Hypergeometric tests were used to quantify the statistical significance of peak overlaps, as implemented in the makeVennDiagram function of the ChIPpeakAnno R package (v3.28.1) [59,60]. The totalTest argument was set to 250,000, as recommended by the authors for stem cell datasets.

### 2.8. Alternative Splicing Quantification

Alternative splicing events were analysed using Whippet v1.6.1 [61] and run on Julia v1.8.0. Bulk RNAseq fastq files were used as input to quantify splicing events using the *whippet-quant.jl* function with the options --stranded, --circ, and --biascorrect, and to generate sam files for visualisation using the --sam option. Differential splicing events were quantified using the *whippet-delta.jl* function with default options. Splicing events were visualised using IGV v2.8.13 [62] after sam files were converted to bam files using samtools v1.9 [53,54] and normalised using the bamCoverage function from deepTools v3.5.2 [63] with the following flags: --normaliseUsing CPM --exactScaling.

## 3. Results

### 3.1. Co-Essential Network Analysis Reinforces CIC’s Role in MAPK Signalling and Identifies Candidate Additional Functions

Genome-wide genetic perturbation screens have been used to map genetic networks, including co-essential networks and genetic interaction networks, to interrogate the functional and protein complex memberships of genes [33]. Co-essential networks refer to genes that exhibit similar fitness patterns when perturbed in genetic screens, and mapping of these coordinated fitness effects (or “co-essentialities”) has been used to group genes into related pathways and to identify novel gene functions in human cell lines [49,64]. Such genetic networks are much larger than protein-protein interaction networks and can capture functional relationships that are complementary to those identified in the latter [33]. Thus, mapping genetic networks can provide an opportunity to decipher complex interactions and functions associated with genes such as *CIC*.

Given that *CIC*’s co-essentiality network has not yet been mapped, we hypothesised that its characterization may extend our insights into CIC function. To identify genes that exhibited co-essentiality with *CIC* across cancer types, we leveraged fitness effect scores—a measurement of the effect of a single gene loss on cell viability [46]—that were calculated from genome-wide KO screens performed by the cancer dependency map (DepMap) project [45,46,47]. We calculated Pearson coefficients between fitness effect scores of *CIC* and 18,333 genes that were perturbed in 739 cancer cell lines and found 88 genes that were co-essential with *CIC* (Pearson coefficient > 0.178 and BH-adjusted *p*-value < 0.05; Figure 1a; Appendix A; Section 2). Given that two genes on the same chromosome (i.e., syntenic gene pairs) may have higher rates of false positives [50], we first focused our attention on the nine co-essential genes that were non-syntenic to *CIC* (i.e., not on chromosome 19; namely, *ATXN1L*, *BECN1*, *DUSP6*, *FXR1*, *KANK1*, *NFYB*, *RASA1*, *SPPL3*, and *ZCCHC12*). Interestingly, many of these genes have roles that have previously been linked to CIC, as described in Table 1. Furthermore, we calculated co-essential genes without syntenic genes and confirmed that these nine non-syntenic candidates indeed remained statistically significant (Appendix A; Section 2).

To complement our efforts to identify genes with fitness effect scores similar to *CIC*, we next identified genes that were co-expressed with *CIC* [65]. To do this, we compared our list of co-essential genes to gene expression correlation profiles from COXPRESdb [48], as performed by Wainberg et al. [50]. The top 50 co-expressed genes (Appendix A; Section 2) included three co-essential genes: *ERF*, *BICRA*, and *PLEKHG2* (ranked 3, 42, and 24, respectively, in the co-essentiality analysis and ranked 3, 19, and 23, respectively, in the co-expression analysis; all located on chromosome 19; Figure 1a; Appendix A). Although it is difficult to determine whether their appearance in both our co-essential screen and as top co-expressed genes is due to an artefact of their genomic proximity, the proteins encoded by these genes have been associated with functions similar to those of non-syntenic genes identified as co-essential with *CIC*, supporting the possibility that CIC does indeed share functionality with these proteins. For example, PLEKHG2 regulates the expression of CDC42, a regulator of cell cycle progression and chromosome segregation [66,67,68]. Notably, *ERF* perturbations had the highest correlation with *CIC*, besides *CIC* with itself, and its expression was also highly correlated with that of *CIC* (z-score = 9.6). *ERF* is located on the same chromosome band as *CIC* (chromosome 19q13.2) and is positioned directly adjacent to and transcribed in the opposite direction of *CIC*. Similar to CIC, ERF is an ETS domain transcriptional repressor and an ERK-dependent kinase that functions as an effector of the RTK signalling pathway [69,70]. Similar to CIC, loss of ERF has been linked to dysregulation of the cell cycle [69], developmental delays, and intellectual disability [71]. ERF and CIC were also recently shown to coordinate to regulate *ETV1* expression in prostate cancer cells [72], further supporting the notion that the two proteins function similarly and, at least in one context, cooperatively. Interestingly given CIC having recently been shown to interact with the SWI/SNF complex to regulate neuronal maturation [32], the co-essential and co-expressed gene *BICRA* encodes the SWI/SNF complex member GLTSCR1 [73], further strengthening functional associations between CIC and the SWI/SNF complex. Given that several of the functions associated with CIC’s co-essential genes are also known to be regulated by MAPK signalling, such as regulation of apoptotic signalling [74], cell cycle progression [75], and cell differentiation [75], these results thus reinforce CIC’s role in the regulation of RTK/MAPK signalling and highlight its potential impact on additional downstream processes that are controlled by this pathway.

**Table 1 cancers-15-02805-t001:** Nine non-syntenic *CIC* co-essential gene candidates and their known functions, and corresponding related functions attributed to CIC.

Candidate Co-Essential Gene	Candidate Gene Function(s)	Related CIC Function(s)
*ATXN1L*	Interacts with and stabilises CIC for DNA binding [29].	Interacts with and is stabilised by ATXN1L [29].
*BECN1*	Regulates apoptosis and cytoskeletal dynamics [76,77]. Mediator of autophagy [76].	No known related function.
*DUSP6*	Negative regulator of MAPK signalling and known target of CIC-mediated repression [21].	Represses DUSP6 expression [21].
*FXR1*	RNA-binding protein, has been implicated in the stabilisation of multiple transcripts through binding to their 5′ or 3′ UTRs [78,79,80].	No known related function.
*KANK1*	Regulates apoptosis and cytoskeletal dynamics [81,82].	No known related function.
*NFYB*	Regulates cell cycle progression, differentiation, and apoptosis [83,84].	Regulates cell cycle progression and differentiation [30].
*RASA1*	Regulates MAPK signalling through suppression of ERK1/2 expression [85]. Regulates cell cycle progression, differentiation, and apoptosis [85].	Regulates MAPK signalling through suppression of ERK1/2 expression [19]. Regulates cell cycle progression and differentiation [30].
*SPPL3*	Protease that participates in the regulation of T-cell responses by interfering with human leukocyte antigen detection [86].	Plays a role in T-cell development [10,87,88,89,90].
*ZCCHC12*	Downstream effector of bone morphogenic protein signalling and has been shown to co-regulate cellular development with AP-1 and CREB [91].	Implicated in cellular development in various contexts [30]. FOS and FOSL1, members of the AP-1 complex, are candidate targets of transcriptional regulation by CIC [22,24].

### 3.2. Lethal Genetic Interactors of CIC Are Associated with Regulation of Apoptotic Processes, Cell Division and Differentiation, and Chromatin Organisation

We further leveraged DepMap data [45,46,47,92,93] to perform in silico genome-wide genetic perturbation screens to identify genes that might genetically interact with *CIC*. Specifically, we screened for genetic perturbations that, in *CIC* mutant cells, resulted in decreases or increases in cell viability relative to *CIC*^WT^ control cell lines (schema shown in Appendix A). Similar to co-essential networks, such genetic interaction networks, including synthetic lethal (SL) and alleviating genetic interactions, can reveal interactions between genes that may indicate shared functions [94,95,96].

Using genome sequencing data from DepMap [92], we identified a non-small cell lung cancer (NSCLC) line (HCC-366) and an endometrial carcinoma line (JHUEM1) with homozygous deleterious alterations predicted to abrogate CIC function (*CIC*^HomDel^; Appendix A–c; Section 2). Notably, the mutation in JHUEM1 is exclusive to the long CIC isoform (CIC-L; Appendix A). This cell line may therefore retain at least some CIC activity encoded by the short CIC isoform (CIC-S). We also identified five additional cancer cell lines with deleterious and potentially trans-heterozygous *CIC* mutations (i.e., two or more mutations, all in regions common to the CIC-S and CIC-L isoforms; *CIC*^T-HetDel^); 53 lines with heterozygous deleterious *CIC* mutations (six of which were exclusive to the CIC-L isoform; *CIC*^HetDel^); and 557 *CIC*^WT^ lines (Appendix A–c; Section 2). To infer the presence of functional CIC in these different cell lines, we used DepMap transcriptome and proteome data [92,93] to analyse *CIC* mRNA and protein abundance and the mRNA abundance of *ETV4* and *ETV5,* which are known targets of CIC transcriptional repression. Compared to *CIC*^WT^ lines, we observed a statistically significant decrease in total *CIC* mRNA levels and in *CIC*-S and *CIC*-L transcript levels only in *CIC*^HetDel^ lines (analysis of variance [ANOVA] followed by Tukey’s Honest Significant Difference [HSD] test, *p*-value < 0.05; Figure 1b; Appendix A). In contrast, we observed a significant decrease in CIC protein levels in *CIC*^HetDel^ and *CIC*^T-HetDel^ lines compared to *CIC*^WT^ lines (ANOVA *p*-value = 0.0021, Tukey’s HSD test *p*-value < 0.01 and < 0.05, respectively; Figure 1b). Protein data were not available for the *CIC*^HomDel^ lines. *CIC*^HomDel^ and *CIC*^T-HetDel^ cell lines both displayed trends towards higher mRNA expression of *ETV4* and *ETV5* compared to *CIC*^WT^ lines (ANOVA *p*-value > 0.05; Appendix A). Given the reduced CIC protein expression observed in *CIC*^T-HetDel^ and *CIC*^HetDel^ lines and the moderate increases in *ETV4* and *ETV5* mRNA levels observed in *CIC*^T-HetDel^ and *CIC*^HomDel^ lines, we deduced that the *CIC*^HomDel^ cell lines likely have reduced or perhaps lack wild-type CIC activity and were therefore candidates for use in in silico genetic screening analyses.

We first conducted an in silico genome-wide screen by comparing the lethality outcomes of 18,322 gene KOs in *CIC*^HomDel^ vs. *CIC*^WT^ cell lines (Appendix A; Section 2). We used a measure called the “lethality probability”, which is provided with DepMap data and which normalises the fitness effect scores to account for variability in screen quality between cell lines [46,47]. This measure is scaled from 0 to 1, where a probability of 0 indicates a non-essential gene (i.e., the gene KO did not affect cell viability) and a probability of 1 indicates an essential gene that was necessary for the survival of that cell line [46,47]. Genes that had significantly different lethality probabilities between *CIC*^HomDel^ and *CIC*^WT^ cell lines (defined as having an adjusted Mann-Whitney U test *p*-value < 0.05 and a median lethality probability > 0.5 in at least one group; Methods) were defined as candidate genetic interactors of *CIC*. We identified 70 such genes, all of which were predicted to be SL interactors of *CIC* (i.e., perturbation of these genes resulted in significantly higher lethality probabilities in *CIC*^HomDel^ lines than in *CIC*^WT^ lines; Figure 1c, Appendix A). We next annotated biological functions related to the predicted SL interactors and summarised these functions into eight distinct groups (Figure 1d; Table 2; Appendix A). Interestingly, some of the pathways associated with *CIC*’s co-essential network, such as the regulation of apoptotic processes, cell cycle, cell division, and cell differentiation, were also associated with the predicted SL interactors of CIC. Notably, we also identified *BECN1* as being both co-essential with and a SL interactor of *CIC* (Appendix A), indicating that CIC likely plays a synergistic role either serially or in parallel with BECN1, and that the combined inactivation of these two genes, at least in the HCC-366 (NSCLC) and JHUEM1 (endometrial carcinoma) cell lines, may not be compatible with cell survival.

Our SL analysis identified *ARID2*, a SWI/SNF complex member [97], as *CIC*’s most significant SL interactor (Figure 1d, Appendix A). This is particularly interesting given that CIC has been shown to function with the SWI/SNF complex to regulate neuronal maturation [32] and that we identified *BICRA*, which encodes another SWI/SNF member, within *CIC*’s co-essential network. Given that the SL interaction with *ARID2* was identified using an NSCLC and an endometrial carcinoma cell line and that the co-essentially network analysis was conducted using 739 pan-cancer cell lines, our data suggests that associations between CIC and SWI/SNF complex members, which have previously been observed in neural progenitor cells [32], may also extend to non-neural contexts.

### 3.3. Nuclear CIC Interacts with Mitotic Regulators and SWI/SNF Complex Members

Our in silico analyses of *CIC*’s genetic interaction network support the notion that CIC appears to play similar roles across cellular contexts (e.g., in MAPK signal transduction, cell development, and regulation of proliferation and cell cycle). We have previously shown that CIC exerts distinct functions depending on its subcellular location (i.e., nuclear versus cytoplasmic) [36]; however, our above analyses did not provide an opportunity to discriminate between functions exerted in different subcellular compartments. To further explore and refine CIC’s functional networks in a nuclear context specifically, and given CIC’s known function as a transcriptional repressor, we next set out to characterise CIC’s nuclear protein interaction network with the assumption that co-immunoprecipitation (co-IP) assays might identify interacting transcriptional regulators. We first performed co-IP assays (Section 2; Figure 2a) against endogenous CIC and N-terminal MYC-fused CIC-S (using an anti-MYC antibody) in nuclear fractions purified from the HEK293A cell line (HEK-*CIC*^WT^). The immunoprecipitates were then analysed using liquid chromatography followed by tandem mass spectrometry (MS; Section 2). This yielded 291 candidate CIC-interacting proteins identified in at least two of four IP-MS replicate experiments (Appendix A), including previously identified CIC protein interactors such as ATXN2L, SMARCA2, SMARCA4, SMARCC1, SMARCC2, and SVIL [29,32]. To identify pathways and protein complexes that were overrepresented in our list of candidate interactors, we performed enrichment analyses and identified mRNA splicing, regulation of RNA metabolism, chromatin remodelling, and chromosome segregation among the significantly enriched processes (BH-adjusted *p*-value < 0.05; Appendix A). To identify candidate interactors that are common to more than one cell line, we next performed IP-MS experiments using nuclear fractions of an immortalised normal human astrocyte (NHA) cell line [98] (hereafter referred to as NHA-*CIC*^WT^). Given CIC’s known function as a transcriptional repressor, we were particularly interested in investigating whether candidate interactions with proteins involved in transcription regulation and chromatin organisation could be replicated. We therefore performed a variation of the IP-MS technique in which ‘trigger peptides’ [99] were used to enhance the sensitivity of the MS for a subset of these proteins of interest (Figure 2a, Appendix A; Appendix A). This approach revealed candidate interactions between CIC and 456 proteins in NHA cells, 156 of which we considered “high confidence” as we observed them in both the HEK and NHA lines (Appendix A). This list included the previously identified interactors ATXN2L, SMARCA2, SMARCC1, and SMARCC2. Analysis of these 156 proteins yielded significantly enriched biological processes (BH-adjusted *p*-value < 0.05) that clustered in 26 groups of terms with similar enrichment memberships, including the groups summarised in Table 3 (see Appendix A for full results; Appendix A). Interestingly, among the proteins involved in mRNA metabolism and processing, we found HNRNPA0 and CDC5L. HNRNPA0 is involved in stabilising 3′ UTRs of mRNA [100], and we also identified the gene encoding this protein as a predicted synthetic lethal interactor of *CIC* (Appendix A). These results raise the possibility that CIC may have a role in the processes listed in Table 3.

Consistent with our earlier observations of *CIC* interacting with members of the SWI/SNF complex (*GLTSCR1* was identified as co-essential and *ARID2* as a SL interaction partner), several members of the SWI/SNF complex were identified among the high-confidence candidate interactors of CIC, including SMARCA2, SMARCC1, SMARCC2, and PBRM1 (Appendix A). ARID1A and ARID2 were also identified as interactors in the NHA line, but not in HEK cells. Mammalian SWI/SNF has broad roles that overlap with those attributed to CIC, such as in transcriptional regulation, cell differentiation, lineage specification, DNA damage response, and RNA splicing [101,102]. Given these links and the fact that CIC and the SWI/SNF complex were recently shown to cooperatively repress the expression of neuronal development genes [32], we wanted to confirm these high-confidence interactions. To do this, we first performed reciprocal IPs by immunoprecipitating SWI/SNF complex members and probing the co-IPs with Western blot assays to detect CIC (Appendix A). We validated the interactions between endogenous CIC and SMARCA2, SMARCC1, ARID1A, and ARID2 in the NHA-*CIC*^WT^ line (Figure 2b–e). We were unable to validate the interaction between CIC and PBRM1. We also performed immunofluorescence (IF) colocalisation assays in *CIC*^KO^ HEK cells stably expressing a FLAG-tagged construct of the short CIC isoform (HEK-*CIC*^KO-D10 + F-CIC-S^) and observed a striking apparent colocalisation between F-CIC-S and ARID1A, ARID2, SMARCA2, and SMARCC1 (Figure 2f,g and Appendix A). Interestingly, SMARCA2, SMARCC1, and ARID1A apparently colocalised with and exhibited similar localisation dynamics to CIC over the entire course of the cell cycle. All four proteins were found throughout the nucleus during interphase but appeared to be excluded from condensed chromosomes at metaphase and telophase. During early cytokinesis, colocalising foci appeared at de-condensing chromosomes and increasingly accumulated until the completion of cytokinesis (Figure 2f,g, arrowheads). Conversely, ARID2 appeared to surround decondensing chromosomes in late telophase and colocalised with CIC only later at early cytokinesis. Although physical interactions between CIC and ARID1A and ARID2 were identified in IP-MS experiments performed in NHA cells but not detected in those performed in HEK cells, as described above, these results indicate the possibility that CIC may also interact, or at least colocalise, with these proteins in HEK cells. Since co-IP assays can also reveal indirect protein associations (e.g., proteins that are part of the same complex but that do not necessarily directly interact [103]), we further characterised these apparent colocalisation patterns using proximity ligation assays (PLAs). PLAs emit a fluorescent signal when antibodies targeting candidate protein interactors are within 40 nm of each other and thus provide a stronger level of evidence for direct interactions [104]. In HEK-*CIC*^KO-D10+F-CIC-S^ cells, PLA signals were compatible with colocalisation between F-CIC-S and ARID1A, SMARCA2, SMARCC1, and ARID2 during interphase (Figure 2h). Notably, the PLA signals were predominantly observed in the nucleus. Overall, these results appear to indicate spatial relationships between CIC and SWI/SNF complex proteins in both HEK and NHA cell lines, raising the possibility that CIC’s functional interaction with the SWI/SNF complex extends beyond the neural progenitor context in which it was first identified [32].

In order to further probe the intriguing patterns of CIC nuclear localisation observed above, we next performed IF assays in the NHA-*CIC*^WT^ line and the HOG line (HOG-*CIC*^WT^) [105], observing similar localisation dynamics in both cell lines as were observed in the HEK cells above (Figure 2i, Appendix A). Ectopic expression of F-CIC-S in HEK-*CIC*^KO2 + F-CIC-S^ cells also displayed similar localisation dynamics as endogenous CIC (Appendix A). To determine if CIC’s distinct dynamic localisation pattern was potentially mediated by mitotic structures, we performed IF colocalisation assays using markers of ɑ-tubulin, CENPA, and AURKB to observe spindles, centromeres, and midbody structures, respectively (Appendix A). CIC did not appear to be associated with spindle structures, centromeres, kinetochore attachments, or midbody structures over the course of the cell cycle.

### 3.4. Loss of CIC Is Associated with Mitotic Defects in Mammalian Cells

While performing the IF assays described above, we observed that the *CIC*^KO^ cells appeared to display a higher frequency of defective mitosis (e.g., improper metaphase alignment, lagging chromosomes) compared to *CIC*^WT^ cells. To investigate this further, additional IF assays in the NHA lines were used to confirm that NHA-*CIC*^KO^ lines had an increased frequency of mitotic defects compared to the parental NHA-*CIC*^WT^ cells (Figure 3a,b). Specifically, the *CIC*^KO^ lines (NHA-*CIC*^KO-A2^ and NHA-*CIC*^KO-H9^) showed 2.2–2.5-fold increases in metaphase alignment defects (two-sided Student’s *t*-test *p*-value < 0.05) and 2–2.3-fold increases in chromosome segregation defects (e.g., lagging chromosomes and micronuclei) at telophase/cytokinesis (*p*-value < 0.01). We also observed a significant increase in metaphase defects in HEK-*CIC*^KO^ cell lines compared to HEK-*CIC*^WT^ cell lines (Appendix A), indicating that the association between CIC loss and increased frequency of mitotic defects is not specific to the NHA-derived lines.

In order to determine if this phenotype extended beyond mammalian cell lines, we performed additional IF assays in brain sections from E13.5 mice with forebrain-specific *Cic* loss (*Cic*^fl/fl^; *FoxG1*^cre/+^) and from heterozygous control mice (*Cic*^fl/+^; *FoxG1*^cre/+^) [106]. Consistent with our in vitro results, *Cic*-null cells in the mitotically active ventricular zone displayed an increased frequency of mitotic defects, including lagging chromosomes and micronuclei (average 2.5-fold increase, two-sided Student’s *t*-test *p*-value < 0.0001; Figure 3c). Together, these observations support a previously unappreciated relationship between CIC loss and mitotic defects in mammalian cells in vitro and in vivo.

### 3.5. Loss of CIC Is Associated with Transcriptional Dysregulation across All Cell Cycle Phases

Given CIC’s dynamic pattern of nuclear localisation and the association between its loss and increased mitotic defects, we hypothesised that CIC’s effects on gene expression regulation may depend on the cell cycle. Since dysregulated expression of dynamically expressed genes may not be detectable in asynchronous bulk cell populations and synchronisation through either chemical or physical means can introduce confounding artefacts [107,108], we therefore performed single-cell RNA-seq (scRNA-seq) on asynchronous NHA-*CIC*^WT^, NHA-*CIC*^KO-A2^, and NHA-*CIC*^KO-H9^ cells. To characterise the transcriptional profiles of CIC-proficient and CIC-deficient cells at different phases of the cell cycle, we first scored cells for defined cell cycle phases (G2, G2/M, M/G1, G2/S, and S phases) [41] using approaches described previously [39,40] (Section 2). For each phase, we then performed a differential expression analysis comparing *CIC*^WT^ and *CIC*^KO^ cells assigned to that phase (Figure 4a and Appendix A), and to explore coordinated patterns in gene expression differences, we also performed a gene set enrichment analysis specific to each phase (Appendix A). The canonical CIC targets *ETV1/4/5* were among the genes found to be significantly more abundantly expressed in *CIC*^KO^ cells at each phase (*ETV4*) or some phases (G1/S for *ETV1*, G1/S, and M/G1 for *ETV5*; BH-adjusted *p*-value < 0.05 and absolute fold-change > 1.5), supporting the validity of our approach. Genes overexpressed in *CIC*^KO^ cells were also consistently found to be enriched for targets of RAS/MAPK signalling (e.g., *ENG*, *ETV1/4/5*, *GNG11*, *GYPC*, *SOX9*), as has previously been shown using bulk transcriptome profiling of CIC-deficient cell lines and tumours [13,20,21,22], indicating that CIC plays a role in the regulation of this signalling cascade at all stages of the cell cycle. In line with CIC’s known role as a transcriptional repressor that interacts with chromatin modifiers such as histone deacetylases and the SWI/SNF complex [22,32], we also observed enrichment of chromatin-related terms within overexpressed genes. Given that our IP-MS analysis identified MECP2 as a high-confidence physical interactor of CIC (Appendix A), of particular interest was the Reactome term “transcriptional regulation by MECP2” (M27862), which was enriched or trended towards enrichment across all five phases (Figure 4b). Genes belonging to this term that were found to be differentially expressed in *CIC*^KO^ cells in at least one phase included *HIPK2*, which encodes a serine-threonine kinase that regulates the activity of a wide range of transcription factors and chromatin regulators [109], and *TNRC6B* and *DGCR8*, both of which are involved in RNA processing [110,111,112]. Other transcriptional activators, such as *ATF4*, *CREB3L2*, *FOSL1*, *MLXIP*, *RXRA*, *SMAD3*, *SOX9/12*, *SREBF1/2*, and *STAT6,* were also found to be more abundantly expressed in *CIC*^KO^ cells, especially at the G1/S and S phases. Conversely, genes encoding proteins involved in transcriptional repression were also identified among those overexpressed in *CIC*^KO^ cells, including, e.g., the polycomb repressive complex 1 (PRC1)-related genes *CBX6* and *PHC2*, the nucleosome remodelling and deacetylase (NuRD) complex members *CHD3*, *MBD3*, and *GATAD2A*, and the histone deacetylase *HDAC7*.

Although elevated expression of target genes in *CIC*^KO^ cells could be a direct consequence of CIC loss given its function as a transcriptional repressor, across all cell cycle phases, approximately half (mean 51%, range 45–60%) of all significantly differentially expressed genes displayed lower expression in *CIC*^KO^ cells (Figure 4a). This is in agreement with results we and others have shown previously from bulk RNA-seq profiling of CIC-proficient and CIC-deficient cell lines and tumour samples [21,29,113]. Interestingly, several gene sets related to cell cycle regulation and checkpoints were enriched for downregulated genes, driven by genes such as *CCNB1* (significantly less abundant in S and G2/M *CIC*^KO^ cells), *CDK1* (G1/S), *MAD2L1* (S and G2/M), *CENPK* (M/G1), and *CCNE2* (G1/S; Figure 4c). *YWHAQ* and *YWHAB*, which are members of the 14-3-3 family of proteins that bind to CIC and inhibit its activity [114] and which also play a role in cell cycle checkpoints and regulation of apoptosis [115], were also found to be underexpressed in *CIC*^KO^ cells (*YWHAQ* reaching statistical significance in all phases except G2/M and *YWHAB* reaching statistical significance in S-phase cells only). Echoing the results of our IP-MS analysis, we also observed an enrichment of genes involved in RNA processing and splicing among genes with lower expression in *CIC*^KO^ cells. These include the small nuclear ribonucleoprotein core protein family members *SNRPB2* and *SNRPD2*, the heterogeneous nuclear ribonucleoproteins *HNRNPH3* and *HNRNPA2B1*, the splicing factor 3B subunit *SF3B4*, and the spliceosome component *PRPF31*, all of which were significantly under-expressed in G1/S-phase *CIC*^KO^ cells. Interestingly, our results also indicated that *NPM1* had significantly lower expression in G2/M-phase *CIC*^KO^ cells, although it showed a similar trend at all phases, especially in the *CIC*^KO-A2^ cell line (Figure 4c). *NPM1* is frequently mutated in acute myeloid leukaemia and encodes a protein involved in functions related to many of the ones mentioned above, including centrosome licensing (and thereby genome stability), ribosome biogenesis, the p53-mediated stress response, and inhibition of cell growth and proliferation [116]. Notably, we also identified NPM1 as a high-confidence physical interactor of CIC (Appendix A).

In order to investigate whether genes that were differentially expressed in *CIC*^KO^ cells might be directly regulated by CIC, we compared our gene list with a set of 150 high-confidence CIC binding sites that our group previously identified in the NHA-*CIC*^WT^ line using chromatin immunoprecipitation followed by sequencing (ChIP-seq) [24]. Of the 135 genes associated with these binding sites (Appendix A), transcripts for 10 of them—*DUSP4*, *DUSP5*, *ETV4*, *ETV5*, *FOSL1*, *HMGA1*, *MAFF*, *OSGIN1*, *TBC1D14*, and *TBC1D22A*—were differentially abundant between NHA-*CIC*^WT^ and NHA-*CIC*^KO^ cells in at least one cell cycle phase in our scRNA-seq data (Appendix A). All of these genes were more highly expressed in NHA-*CIC*^KO^ cells, which is consistent with CIC’s known function as a transcriptional repressor. Indeed, *ETV4* and *ETV5* are well-established targets of CIC transcriptional regulation, and *DUSP4*, *DUSP5*, *FOSL1*, and *MAFF* have also previously been proposed to be direct targets of CIC in different cellular contexts (Table 4). Several of these genes are also involved in MAPK signalling (Table 4), which is consistent with CIC’s known function within this pathway.

Overall, these results indicate that CIC appears to regulate the expression of genes involved in a wide range of functions, including MAPK signalling, transcriptional regulation, mitotic checkpoint activity, and RNA processing. Notably, our results also indicate that CIC regulates the expression of similar groups of genes across different phases of the cell cycle. Furthermore, candidate direct targets of CIC transcriptional regulation appear enriched for genes involved in functions that we have repeatedly identified in this study, including MAPK signalling, AP-1 activity, and apoptosis, further implicating CIC activity within these functional groups.

### 3.6. CIC Binding Sites Overlap with a Subset of SWI/SNF Binding Sites

In light of our observations of genetic and physical interactions between CIC and members of the SWI/SNF complex and of a recent study showing that CIC functions with the SWI/SNF complex to regulate gene expression in the context of neuronal development [32], we set out to explore whether CIC may function cooperatively with the SWI/SNF complex to regulate gene expression in contexts beyond the one investigated by Hwang et al. [32]. To do this, we obtained publicly available ChIP-seq data for CIC [22] and ARID1A, BRD4, BRD9, and SMARCA4 [51] generated from mouse embryonic stem cell (mESC) cultures. We chose these datasets to limit, as much as possible, biological variability between cell lines since we were unable to find SWI/SNF ChIP-seq data from human cell lines in which CIC ChIP-seq has been performed. Supporting the translatability of human and mouse datasets, 25 orthologous gene pairs were associated with CIC peaks in both the NHA dataset [24] and the mESC dataset [22] (Appendix A), including the established CIC targets *CCND1/Ccnd1*, *DUSP4/Dusp4*, *ETV4/Etv4*, *ETV5/Etv5*, *GPR3/Gpr3*, *LRP8/Lrp8*, *PTPN9/Ptpn9*, *SPRED2/Spred2*, and *SPRY4/Spry4*. Of the 108 peaks identified in the mESC CIC ChIP-seq dataset (see Section 2), 26 overlapped with ARID1A peaks (hypergeometric test *p* = 1.38 × 10^−12^), 60 with BRD9 peaks (*p* = 1.71 × 10^−40^), 84 with BRD4 peaks (*p* = 2.26 × 10^−38^), and 102 with SMARCA4 peaks (*p* = 8.66 × 10^−32^). Notably, of the 100 genes associated with these 108 mESC CIC peaks, the human orthologues for 10 of them were found to be differentially expressed in at least one cell cycle phase in our scRNA-seq data (*DDX54*, *DUSP4*, *DUSP5*, *ETV1*, *ETV4*, *ETV5*, *HSPG2*, *MAFF*, *SREBF2*, and *TTLL12*). The 14 peaks associated with these 10 genes all overlapped with SWI/SNF peaks (14/14, 13/14, 10/10, and 4/14 sites overlapped with SMARCA4, BRD4, BRD9, and ARID1A peaks, respectively; Appendix A). Together, these results indicate the possibility that CIC may indeed function with the SWI/SNF complex to regulate transcription.

### 3.7. CIC Loss Destabilises Splicing at Untranslated Regions

Given our observations that RNA splicing factors share genetic and proteomic interactions with CIC and that RNA splicing genes are dysregulated in CIC-deficient cell lines, we posited that loss of CIC may have a global effect on transcript splicing. To investigate this, we applied Whippet, a tool for de novo detection and quantification of alternative splicing events using Shannon entropy-based methods [61], to bulk RNA-seq data we generated previously from three biological replicates of each of the NHA-*CIC*^WT^, NHA-*CIC*^KO-A2^, and NHA-*CIC*^KO-H9^ lines [24] (GEO accession: GSM5708721). We detected 29,020, 28,104, and 27,042 alternative splicing events in at least two replicates of the NHA-*CIC*^WT^, NHA-*CIC*^KO-A2^, and NHA-*CIC*^KO-H9^ cell lines, respectively (Appendix A). Whippet attributes entropy scores to genes identified as undergoing alternative splicing, with scores ranging from 0 to log_2_(number of isoforms). We did not observe differences in the overall distribution of entropy scores between the NHA-*CIC*^WT^ line and either the NHA-*CIC*^KO-A2^ or NHA-*CIC*^KO-H9^ lines (Kolmogorov–Smirnov test *p*-value > 0.05 in both comparisons; Figure 5a), nor did we observe a significant difference in the proportion of high-entropy events (entropy scores > 1.5; i.e., more than three expressed isoforms) between the NHA-*CIC*^WT^ and NHA-*CIC*^KO^ cell lines (Appendix A). Notably, however, the proportion of high-entropy events identified in the NHA cell lines (~2%; Appendix A) was higher than those observed for hepatocellular carcinoma samples, which showed ~0.10% and ~0.12% high-entropy events in control and tumour samples, respectively [61]. This is consistent with previous reports showing that brain cells have higher isoform complexity than other tissue types [124]. Together, these results indicate that CIC loss does not appear to result in marked global changes in alternative splicing events, at least in NHA cells.

Additionally, to determine whether CIC loss affected specific types of splicing events, we used Whippet to perform differential usage analysis and categorised the events by splicing type, comparing the parental NHA-*CIC*^WT^ line to both NHA-*CIC*^KO^ lines. We identified 787 significant differential splicing events affecting 468 genes (absolute delta Ψ > 0.1 and probability > 0.9, as described by Georgakopoulos-Soares et al. [125]; Figure 5b; Appendix A). Interestingly, of the eight different types of splicing events that Whippet can detect—namely alternative acceptor, alternative donor, alternative first exon, alternative last exon, core exon, retained intron, tandem transcription end sites, and tandem transcription start sites (TSSs) [61]—the splicing events that were found to be differential between *CIC*^WT^ and *CIC*^KO^ cells were all categorised as one of only three types: transcriptional end sites (or tandem polyadenylated [poly-A] sites; 589 events affecting 360 genes), TSSs (or tandem 5′UTR; 195 events affecting 109 genes), and retained introns (three events affecting three genes; Figure 5b; Appendix A; Appendix A). Furthermore, we observed more genes containing events that led to a shorter 3′ UTR (i.e., usage of an earlier poly-A site; ~200 genes) compared to those that lengthened it (~150 genes) in the NHA-*CIC*^KO^ cell lines (Figure 5c). Conversely, tandem TSS events were equally distributed between those that lengthened or shortened the 5′ UTR. These results may indicate that loss of CIC can affect 3′ end usage and, to a lesser extent, 5′ end usage, and that it may favour the shortening of 3′ UTRs. To investigate the genes affected by differential splicing events upon CIC loss, we annotated these genes and summarised their functions into 18 categories (Appendix A; Appendix A). We found genes associated with roles such as regulation of vesicle transport, apoptosis and autophagy, cell development and differentiation, the microtubule cytoskeleton, the cell cycle, DNA repair, and chromosome segregation (Appendix A). Interestingly, the SWI/SNF complex members *ARID1A* and *ARID1B* both displayed alternative TSS usage in *CIC*^KO^ cells (Appendix A). The differential intron retention events affected the ribosomal proteins *RPL7A* and *RPL35A* (exclusion in the *CIC*^KO^ cells; Figure 5d,e) and the ubiquitin conjugate enzyme *UBE2N* (inclusion; Figure 5f), which is involved in DNA repair [126]. Differential 3′ UTR usage was detected in *BCL2L11*, which encodes the proapoptotic BH3-only protein BIM, resulting in a shortening of this region in *CIC*^KO^ cells. Interestingly, in rat neurons, ERK signalling has been shown to regulate *Bim* expression via the 3′ UTR [127]. Given that 3′ UTR length can affect the stability, localisation, and translation of mRNA [128], these results suggest that loss of CIC may affect the functions of these processes through dysregulation of transcript splicing.

Given that UTR length and intron retention events have both been shown to affect gene expression [129,130,131], we next compared the list of genes affected by significant alternative splicing events to those that were found to be differentially expressed in our scRNA-seq data. We identified 51 genes affected by alternative splicing that were also significantly differentially expressed (31 and 20 genes upregulated and downregulated in NHA-*CIC*^KO^ cells, respectively; Appendix A). Consistent with the notion that intron retention can downregulate gene expression [131], *RPL7A*, which was found to exclude an intron in the NHA-*CIC*^KO^ cell lines that was retained in the NHA-*CIC*^WT^ line (Figure 5d), showed higher expression in NHA-*CIC*^KO^ lines at G1/S, S, and M/G1 phases, whereas *UBE2N*, which retained an intron in *CIC*^KO^ cells (Figure 5f), was significantly downregulated in G1/S-phase *CIC*^KO^ cells (Appendix A). We also observed several genes associated with cell cycle regulation, namely *CCNC* [132], *HMGA2* [133], and *PLAC9* [134], that had differential UTR events as well as differential gene expression (Figure 5i–k and Appendix A). Interestingly, *NRG1*, which encodes the NRG1 ligand that binds to ERBB receptors to activate MAPK and other signalling pathways [135] and which is also involved in neuronal development [136], was also found to include an additional TSS in *CIC*^KO^ cells (Figure 5l) and was overexpressed in G1/S-phase *CIC*^KO^ cells (Appendix A). *NRG1* gives rise to >30 isoforms, which are grouped into six types (I-VI) [137]. The alternative TSS identified in *CIC*^KO^ cells affects NRG1 isoforms 3, 6, 7, and 8 (Uniprot ID Q02297-3, Q02297-6, Q02297-7, and Q02297-8, respectively), which share a common TSS and which belong to types II (isoform 3, predominantly expressed in mesenchymal cells) and III (isoforms 6-8, the major isoforms expressed in neurons) [136]. Since type III NRG1 isoforms bind ERBB2, ERBB3, and ERBB4 [135,137], dysregulation of type III genes may lead to dysregulation of MAPK and other signalling pathways. Altogether, our profile of the alternative splicing patterns in NHA cell lines shows that loss of CIC is associated with destabilised splicing predominantly at UTRs. A subset of genes affected by alternative splicing also show dysregulated gene expression, including genes associated with roles that were previously known or have emerged as part of this study as being associated with CIC, such as MAPK signalling, DNA repair, cell cycle regulation, and neuronal development.

## 4. Discussion

In this study, we used bioinformatics, mass spectrometry, genomics, single-cell RNA-seq, and microscopy techniques to expand our understanding of CIC function in human and mouse cells and tissues. Results from these orthogonal approaches converged to implicate CIC in cell cycle regulation and splicing in addition to its known function as an effector of MAPK signalling (see Appendix A for a summary of genes/proteins that were identified as part of CIC’s functional networks using different ‘omics approaches). Furthermore, we uncovered multiple lines of evidence indicating that CIC interacts with the SWI/SNF complex in additional cellular contexts than the neuronal progenitor one where the interaction was first observed [32] and that CIC may rely on this interaction for some of its functions in these novel cellular contexts. Strikingly, we also observed that CIC loss was associated with an increased frequency of mitotic defects in human cell lines and in an in vivo mouse model of forebrain-specific CIC loss.

Hwang et al. [32] recently showed that CIC functions with the SWI/SNF complex to regulate neuronal differentiation. Our results support and expand these observations into additional cellular contexts, notably demonstrating, for the first time, that CIC physically interacts with the SWI/SNF complex members SMARCA2, SMARCC1, ARID1A, and ARID2 in NHA and HEK cell lines. Additionally, we found that CIC co-localised with ARID1A, SMARCA2, and SMARCC1 throughout the cell cycle, displaying a dynamic relocalisation pattern that included foci formation at de-condensing chromosomes during initiation of nuclear envelope reassembly and accumulation of these foci until completion of cytokinesis.

Our co-essentiality and co-expression analyses indicated that CIC may share functionality with BICRA/GLTSCR1, a component of the recently identified non-canonical GBAF SWI/SNF complex [73]. Our analysis of publicly available ChIP-seq data from mESCs [22,51] also provided further support for a potential association between CIC and the GBAF sub-complex. Specifically, we found that CIC binding sites overlapped with those of BRD4, which has been shown to interact with BICRA/GLTSCR1 [73,138] and to recruit the GBAF complex to naive pluripotency genes in mESCs [51], and those of BRD9, which is a required component of the GBAF complex and an optional component of the npBAF and nBAF complexes [139]. Although the nature of the GBAF complex’s roles in mammalian development and disease remain poorly understood, early evidence is consistent with the notion that it is a key mediator of these processes [139], warranting further investigation into CIC’s potential association with this non-canonical sub-complex. Given that CIC’s transcriptional repressor function was found to be dependent on its interaction with the SWI/SNF complex during neuronal development [32], together, our observations indicate the possibility that this cooperative relationship could be at play in additional cellular contexts and may rely on interactions with one or more sub-complexes.

In addition to CIC’s interactions with SWI/SNF complexes, another major finding of our study is that CIC appears to play a role in mitotic integrity, as evidenced by the observation that its loss is associated with increased mitotic defects in several in vitro and in vivo models. Notably, these two observations may be related to one another; indeed, the SWI/SNF complex has been implicated in the maintenance of genomic stability, and disruption of its activity has been associated with mitotic defects similar to the ones we observed in this study, such as faulty chromosome segregation [140,141,142,143]. The mapping of CIC’s genetic and physical interaction networks also revealed several additional interactors involved in cell cycle and mitotic control. For example, *BECN1*, which was predicted to share both co-essentiality and a synthetic lethal partnership with *CIC*, has been shown to promote accurate kinetochore anchoring to mitotic spindles [77], proper cytokinetic abscission [144], and to regulate autophagy and apoptotic processes [76]. Notably, inhibition of *BECN1* expression is associated with phenotypes similar to those we observed in CIC-deficient cells, such as an increased frequency of cytokinesis defects [77]. BECN1’s known role in supporting the formation of mitotic structures, together with our failure to observe co-localisation between CIC and these structures in mitotic cells, raise the possibility that CIC may be involved in a parallel role that converges with BECN1’s function in maintaining mitotic fidelity, which could explain why their combined loss is predicted to be incompatible with cell survival.

Another important finding that emerged from our study is the observation that CIC interacts (both physically and genetically) with a wide range of proteins and genes involved in RNA processing, splicing, and mRNA stability. For example, the list of 156 proteins we identified as high-confidence candidate physical interactors of CIC included at least 38 proteins annotated as being involved in functions related to RNA splicing, rRNA/mRNA metabolism and processing, and ribosome biogenesis, which is consistent with the notion that CIC physically interacts with RNA processing complexes. Furthermore, genetic interactions were also observed between *CIC* and genes that play a role in mRNA stability, such as *FXR1*, which has been shown to stabilise transcripts through binding to their 5′ or 3′ UTRs [78,79,80] and which was identified as co-essential with *CIC*, and *HNRNPA0*, which can also stabilise transcripts through binding to their 3′ UTRs [100] and which was identified as a synthetic lethal partner of *CIC* as well as a high-confidence physical interactor (Appendix A). Through analysis of bulk RNA-seq data that our group previously generated from *CIC*^WT^ and *CIC*^KO^ NHA cell lines, we found that loss of CIC is associated with dysregulated at 5′ and 3′ UTRs, raising the possibility that CIC also plays a role in mRNA stability.

It is worth noting that, in addition to the putative novel roles discussed above, our study also reinforced CIC’s known role as an effector of MAPK signalling and provided clues as to interactions between this established function and CIC’s possible novel ones. For example, *CIC*’s co-essential network, which is composed of genes with which *CIC* is likely to share functionality, included several known regulators of MAPK signalling, including *DUSP6*, *RASA1*, and *ERF*. Notably, CIC and ERF have been shown to co-regulate the expression of *ETV1* in prostate cancer, where their combined loss and the subsequent increase in *ETV1* expression contribute to disease progression [72]. As mentioned above, the two genes, which are located directly adjacent to one another on chromosome 19q, are co-deleted in 1p/19q co-deleted low-grade gliomas and also frequently in prostate adenocarcinomas (~10–12% of cases) [72], suggesting that the dual loss of CIC and ERF (possibly with BICRA/GLTSCR1, as discussed above) may favour tumour development, possibly through their combined dysregulation of MAPK signalling. Further studies aimed at distinguishing the joint and distinct roles of CIC and ERF are thus warranted in order to reveal the extent and implication of their cooperative roles.

While our multi-omic analysis uncovered several putative CIC functions, we acknowledge that there are limitations to such analyses. For example, both genetic interaction networks [145] and protein interaction networks [146] are known to contain context-dependent interactions. To focus on generalised CIC functions, we used several methods to mitigate this possible limitation; for example, we used multiple contexts when mapping both genetic interactors (*CIC* mutant lines included a lung cancer and an endometrial carcinoma line; *CIC* wild-type lines included 557 lines from 29 cancer contexts) and protein interactors (NHA and HEK cell lines). We also highlighted not only interacting genes/proteins but also the broad functional pathways that were enriched in multiple assays (e.g., MAPK signalling, cell cycle regulation, SWI/SNF function), which are known to be more conserved [147]. While this approach may miss context-dependent interactions, we argue that multi-omic and multi-context studies such as this one are crucial to expanding our understanding of the complexities of CIC functions. Our study thus provides valuable insights into CIC’s roles and provides multiple avenues for future, more focused studies.

## 5. Conclusions

Overall, our study provides a rigorous exploration of CIC’s genetic and proteomic interaction networks and considerably expands our understanding of the diverse cellular processes in which CIC appears to play a role. Beyond the insights it provides into CIC function specifically, this work also presents a model for how multi-omic, network-based analyses that leverage both public and novel datasets can reveal novel and interconnected roles for pleiotropic genes/proteins across cellular and subcellular contexts. For instance, our results raise the intriguing possibility that CIC may be required to ensure faithful mitotic division and that this function may be related to its interactions with SWI/SNF complexes and/or with RNA processing and spliceosomal complexes. This work thus introduces novel avenues through which CIC can act as a pan-cancer tumour suppressor, and we anticipate that it will catalyse further innovative investigations into this intriguing protein’s varied roles.

## Figures and Tables

**Figure 1 cancers-15-02805-f001:**
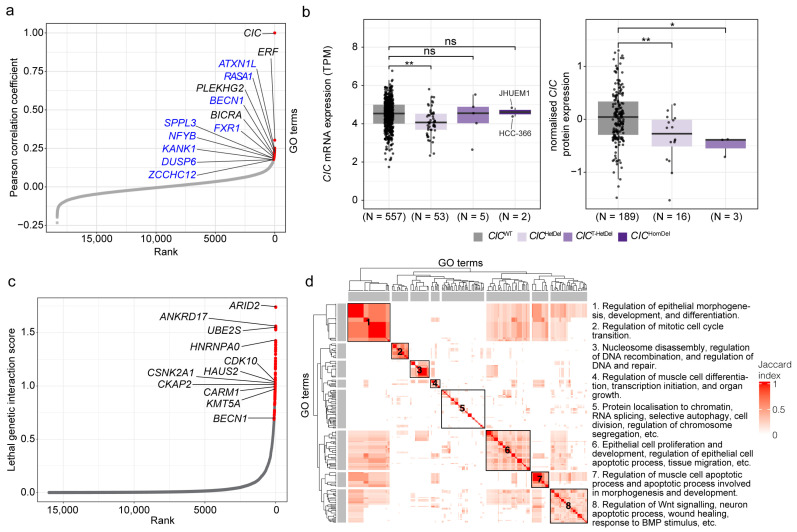
*CIC* is predicted to share co-essentialities and synthetic lethal interactions with regulators of cell proliferation and differentiation and chromatin remodelers. (**a**) Pearson correlation coefficients between the lethality probabilities of *CIC* and those of all genes targeted by DepMap genetic screens (*n* = 18,333) across 739 cancer cell lines. Red points indicate genes with coefficients above the inflection point (88 genes; coefficient > 0.178). Non-syntenic genes are labelled in blue, and selected syntenic genes are labelled in black. (**b**) Transcriptomic and proteomic data from DepMap were compared across cancer cell lines. **Left**: *CIC* mRNA abundance in *CIC*^HomDel^ (*n* = 2), *CIC*^T-HetDel^ (*n* = 5), *CIC*^HetDel^ (*n* = 53), and *CIC*^WT^ (*n* = 557) cancer cell lines. mRNA expression values were normalised to transcripts per million (TPM). Analysis of variance (ANOVA) *p*-value = 0.00407. **Right**: CIC protein expression (scaled and mean-centred) in *CIC*^T-HetDel^ (*n* = 3), *CIC*^HetDel^ (*n* = 16), and *CIC*^WT^ (*n* = 189) cancer cell lines. ANOVA (*p*-value = 0.00217) followed by Tukey’s HSD analysis: * *p*-value < 0.05, ** < 0.01. Not significant (ns; *p*-value > 0.05). (**c**) Ranked lethal genetic interaction scores of *CIC*. Each point represents the interaction score of a single gene KO compared between the *CIC*^HomDel^ (*n* = 2) and *CIC*^WT^ (*n* = 557) cancer cell line groups, where a higher interaction score indicates an increased likelihood of a gene KO being lethal specifically in the *CIC*^HomDel^ cancer cell line group (see Section 2). Red points indicate significant lethal genetic interactors of *CIC* (adjusted Mann-Whitney U test *p*-value < 0.05 and median lethality probability >0.5 in at least one cell line group). (**d**) Heatmap showing Jaccard index similarities between 232 GO terms enriched for the SL partners of *CIC* (**left**). Jaccard index-based hierarchical clustering was used to summarise the GO terms into eight distinct functional groups, and these functions were summarised in the text (**right**; see also Appendix A).

**Figure 2 cancers-15-02805-f002:**
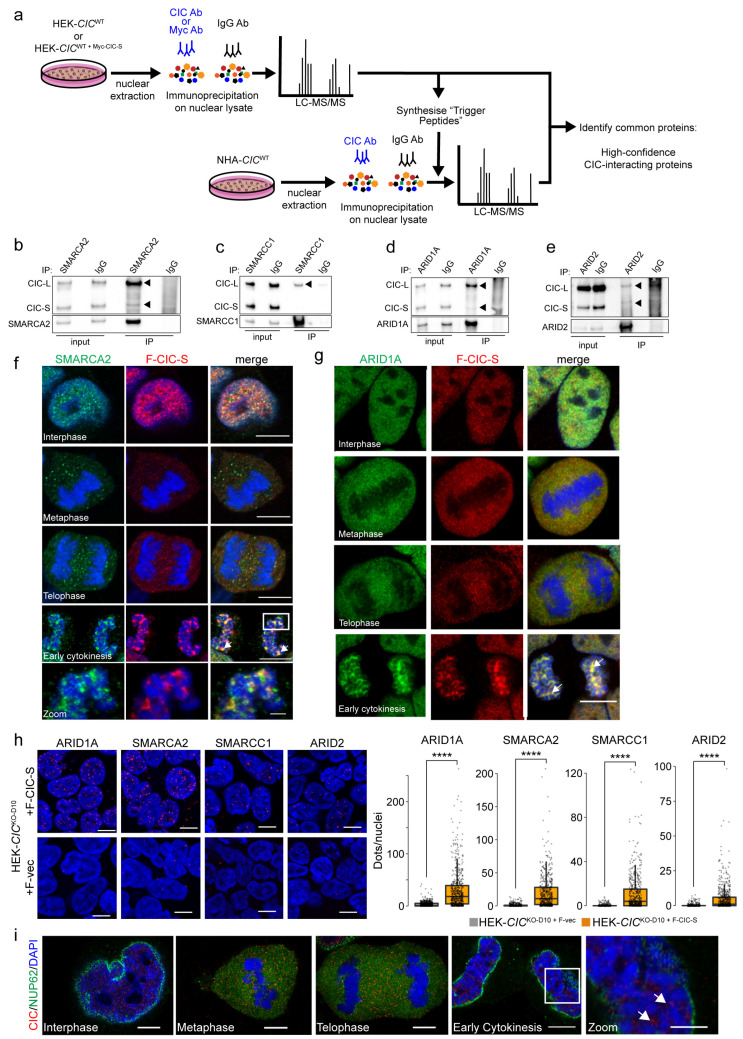
**Figure 2**. CIC interacts with members of the SWI/SNF complex. (**a**) Workflow of CIC IP-MS in HEK-*CIC*^WT^ and NHA-*CIC*^WT^ cell lines. See Section 2 and Appendix A for more detail. (**b**–**e**) Reciprocal IPs confirm interactions between CIC and select SWI/SNF subunits. SMARCA2 (**b**), SMARCC1 (**c**), ARID1A (**d**), and ARID2 (**e**) were immunoprecipitated from nuclear lysates of the parental NHA-*CIC*^WT^ cell line, and CIC interaction was visualised using western blotting. Input lanes indicate nuclear lysate samples used as input for the respective IPs. The CIC-L and CIC-S bands are marked with arrowheads. ((**f**),**g**) Localisation of F-CIC-S (FLAG, red), SMARCA2 (**f**, green), and ARID1A ((**g**), green) in HEK-*CIC*^KO-D10 + F-CIC-S^ cells at indicated phases of the cell cycle, visualised using IF. DNA was stained with DAPI (blue). Arrows indicate colocalisation of CIC and the relevant interactor at early cytokinesis (yellow foci). Scale bars: 10 µm and 5 µm (zoomed image in (**f**)). (**h**) **Left**: representative PLAs using antibodies against FLAG and ARID1A, SMARCA2, SMARCC1, or ARID2 in HEK-*CIC*^KO-D10^ cells expressing F-CIC-S (HEK-*CIC*^KO-D10 + F-CIC-S^, top) or an empty vector (HEK-*CIC*^KO-D10 + F-Vec^, bottom). DNA was visualised using DAPI staining (blue). Red spots are PLA signals indicating proximity of the proteins assayed (<40 nm). **Right**: Tukey boxplots showing quantifications of PLA spots per nuclear region. ARID1A, *n* = 732 and 583; SMARCA2, *n* = 564 and 533; SMARCC1, *n* = 564 and 533; ARID2, *n* = 564 and 588 (nuclei counts for HEK-*CIC*^KO-D10 + F-Vec^ and HEK-*CIC*^KO-D10 + F-CIC-S^, respectively). **** *p*-value < 0.0001 (Welch’s *t*-test). (**i**) IF staining of endogenous CIC (red) and the nuclear envelope protein NUP62 (green) in the parental NHA-*CIC*^WT^ line. CIC shows a punctate localisation pattern (arrows) throughout the decondensing nucleus. Scale bars: 10 µm and 5 µm (zoomed image).

**Figure 3 cancers-15-02805-f003:**
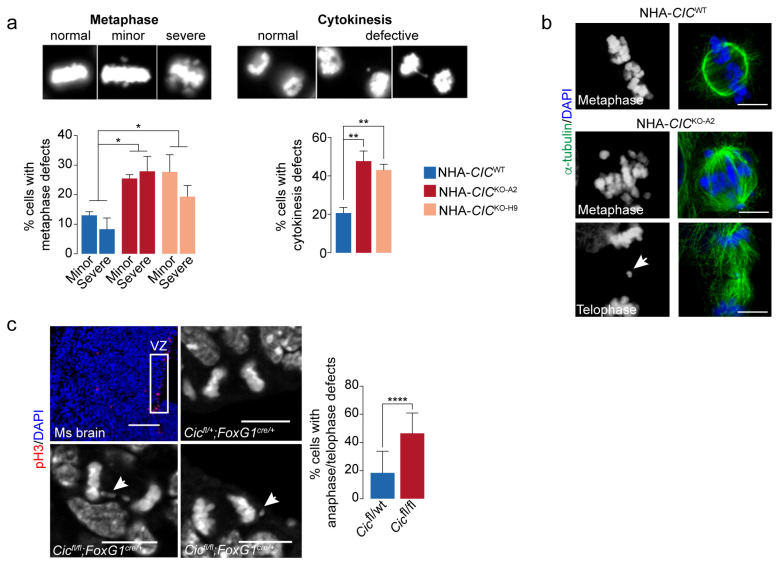
Cells lacking CIC have an increased frequency of mitotic defects. (**a**) **Top**: representative images of NHA cells with normal or defective metaphase and cytokinesis. **Bottom**: proportions of cells from indicated cell lines assigned to each category. Bars represent the mean from three independent experiments, and error bars indicate the standard error of the mean (s.e.m.). NHA-*CIC*^WT^, *n* = 146 and 122; NHA-*CIC*^KO-A2^, *n* = 99 and 94; NHA-*CIC*^KO-H9^, *n* = 104 and 89 (metaphase and cytokinesis cell counts, respectively). (**b**) Representative images of an NHA-*CIC*^WT^ cell with normal metaphase (**top**) and NHA-*CIC*^KO-A2^ cells with defective metaphase alignment (**middle**) and a lagging chromosome at telophase (arrowhead, **bottom**). DNA was stained with DAPI (blue) and microtubules were visualised using α-tubulin staining (green). **Left**: DAPI staining alone. Scale bars: 10 µm. (**c**) **Left**: mitotically active (boxed) ventricular zone (VZ) region from an E13.5 Cic^fl/+^;FoxG1^cre/+^ mouse forebrain and representative images of a normal cell division in a *Cic*^fl/+^; *FoxG1*^cre/+^ mouse (**top right**) and a lagging chromosome and a micronucleus in *Cic*^fl/fl^; *FoxG1*^cre/+^ mice (arrowheads, **bottom left** and **right**, respectively). DNA was stained with DAPI (blue) and mitotically active cells were visualised using pH3 staining (red). **Right**: proportions of cells with defective anaphase/telophase. Bars represent the mean from four animals for each genotype, and error bars indicate s.e.m (*n* = 186 cells for *Cic*^fl/+^; *FoxG1*^cre/+^ mice and 180 for *Cic*^fl/fl^; *FoxG1*^cre/+^ mice). Scale bars: 100 µm (**top left**) and 10 µm. * *p*-value < 0.05, ** < 0.01, **** < 0.0001 (two-sided Student’s *t*-test (**a**,**c**)).

**Figure 4 cancers-15-02805-f004:**
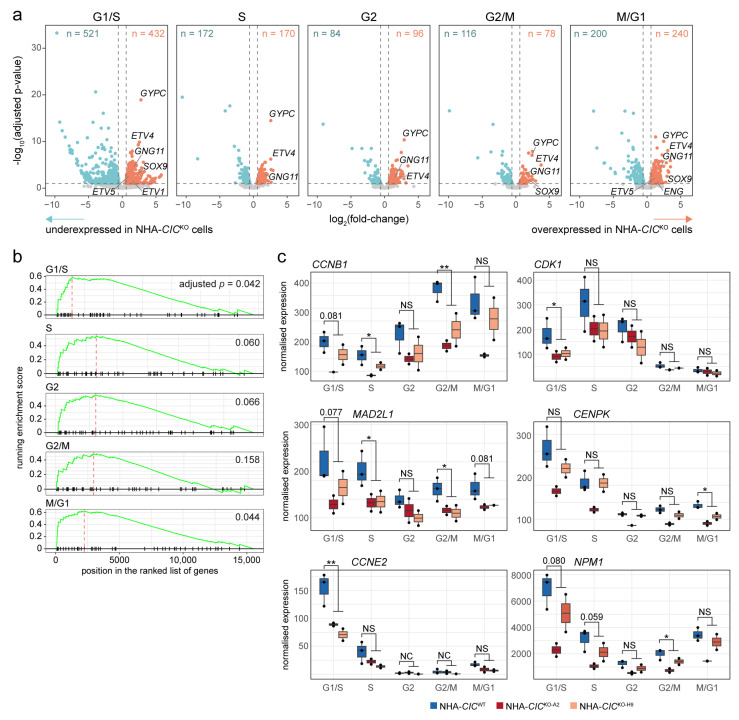
*CIC*^KO^ cells display dysregulated expression of genes involved in chromatin organisation and cell cycle checkpoints. (**a**) Volcano plots depicting differential expression analysis results for *CIC*^KO^ versus *CIC*^WT^ cells assigned to each indicated phase. Genes considered to be significantly differentially expressed (BH-adjusted *p*-value < 0.05 and absolute fold-change > 1.5) are coloured, and their numbers are indicated at the top of each plot. (**b**) Graphical representation of GSEA results for the Reactome term “transcriptional regulation by MECP2”. (**c**) Normalised expression of indicated genes across cell lines and phases. * BH-adjusted *p*-value < 0.05, ** < 0.01 (Wald test as implemented by DESeq2), NS: non-significant, NC: adjusted *p*-value not calculated due to low expression (see Section 2).

**Figure 5 cancers-15-02805-f005:**
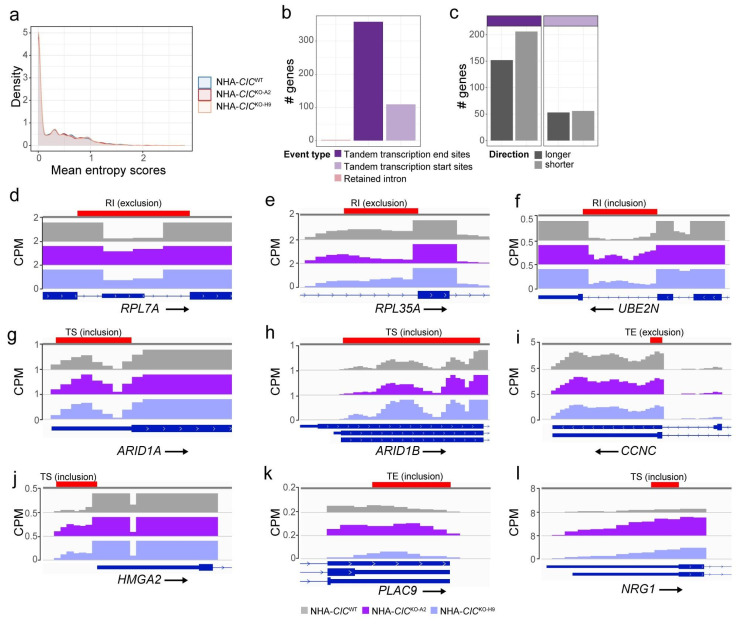
*CIC*^KO^ cells show differential splicing predominantly at UTRs. (**a**) Density plot showing the proportion of genes with different entropy scores in NHA cell lines. Entropy scores are shown as the mean of three replicates. Kolmogorov–Smirnov tests between the distributions of the NHA-*CIC*^WT^ line and both NHA-*CIC*^KO^ lines were not significant (*p*-value > 0.05). (**b**) The number of genes affected by significant differential splicing events (absolute delta Ψ > 0.1 and probability > 0.9) that were detected when comparing the NHA-*CIC*^WT^ line to both NHA-*CIC*^KO^ lines combined, grouped into the indicated categories (see Appendix A for all detected events). (**c**) The number of genes with alternative transcriptional start or end site events from panel b that led to either longer or shorter UTR sites. See panel b for the event type colour legend. (**d**–**l**) Representative images from IGV of one replicate each of the NHA-*CIC*^WT^, NHA-*CIC*^KO-A2^, and NHA-*CIC*^KO-H9^ lines showing genes affected by significant differential splicing events. Reads were normalised to counts per million (CPM) for visualisation. Red bars highlight the genomic regions where the events were detected, and the event types are indicated above (retained intron [RI], tandem transcription end site [TE], and tandem transcription start site [TS]). RefSeq transcript models are shown in dark blue below the sample tracks, with transcriptional direction indicated by arrows.

**Table 2 cancers-15-02805-t002:** Summary of biological functions enriched in candidate SL interactors (see Appendix A for details).

Enriched Functions	Selected SL Interactors in Relevant Term(s)
Regulation of epithelial cell function (morphogenesis, differentiation, proliferation, and migration)	*RREB1, GATA3, PAX8, CEBPB,* and *SP1*
Regulation of cell cycle processes and cell division	*ANKRD17, CARM1, CSNK2A1, HAUS2, CDK10, BECN1, UBE2S, CKAP2,* and *KMT5A*
Chromatin/nucleosome disassembly	*ARID2*, and *NFE2*
Histone modification and chromatin organisation	*BECN1, CARM1, GATA3, KMT5A,* and *RIF1*
Regulation of double-stranded break repair	*ARID2, RIF1,* and *ATP23*
RNA splicing	*LUC7L2, HNRNPA0,* and *RBM23*
Apoptotic processes	*CEBPB, GATA3, ILK, NKX2-5, PAX8,* and *PRKC1*
Wnt signalling	*FERMT2, ILK, GATA3, GNAQ, CSNK2A1, NKX2-5,* and *CALCOCO1*

**Table 3 cancers-15-02805-t003:** Summary of selected biological functions enriched in high-confidence CIC protein interactors (see Appendix A for details).

Enriched Functions	High-Confidence Interactors
RNA splicing and ribosomal and messenger RNA (rRNA and mRNA, respectively) metabolism and processing	>30 interactors
DNA conformation change and duplex unwinding	DDX3X, G3BP1, HNRNPA2B1, HP1BP3, NPM1, RUVBL1, TOP1, TTN, and XRCC5/6
Chromatin remodelling and nucleosome organisation	ACTB, HNRNPC, HP1BP3, NPM1, PBRM1, RUVBL1, SMARCA2/A4/A5/C1/C2, TOP1, and TRIM28
Chromatin organisation and epigenetic regulation of gene expression	ACTB, DDX21, FMR1, GLYR1, HNRNPU, MECP2, MKI67, SIN3A, SMARCA5, SNW1, TRIM28, and UPF1
Telomere organisation and maintenance	DKC1, GAR1, GNL3, HNRNPA1/A2B1/C/D/U, NAT10, UPF1, XRCC5/6, and YLPM1
Intrinsic apoptotic signalling	DDX3X, DDX5, DNM1L, HNRNPK, NONO, PRKDC, RPL11, SFPQ, and SNW1

**Table 4 cancers-15-02805-t004:** Summarised functions of selected genes that are bound by CIC (ChIP-seq) and that show differential expression between *CIC*^WT^ and *CIC*^KO^ cells (scRNA-seq).

Gene(s)	Function(s)
*ETV4* and *ETV5*	Effectors of MAPK signalling. Known targets of CIC-mediated transcriptional regulation in multiple cellular contexts [19]
*DUSP4* and *DUSP5*	Negative feedback regulators of MAPK signalling [117]. Proposed direct targets of CIC in HEK cells, mouse embryonic stem cells, and developing mouse brains [21,22,32].
*FOSL1*	Proto-oncogene and member of the FOS gene family, which encodes a group of proteins that can dimerize with proteins of the JUN family to form the transcription factor AP-1. Activated by the RAS/MAPK signalling cascade through phosphorylation by ERK2, ERK5, and RSK2 [118]. Proposed direct target of CIC in mouse embryonic stem cells [22].
*MAFF*	Member of the basic leucine zipper (bZIP) family of transcription factors. Can also participate in the formation and binding activity of the AP-1 complex [119]. Has been associated with a variety of functions, mostly related to the stress response. Its activity has been implicated in cancer [120,121]. Has been implicated as a feedback regulator of the MAPK signalling cascade, potentially downstream of AP-1 activity [122]. Proposed direct target of CIC in mouse embryonic stem cells [22].
*OSGIN1*	Involved in oxidative stress and DNA damage responses. Tumour suppressor that acts downstream of and in concert with p53 to induce mitochondrial cytochrome c release and apoptosis [123].

## Data Availability

scRNA-seq data have been deposited in the Gene Expression Omnibus (GEO; accession GSE193765). IP-MS data have been deposited with the ProteomeXchange Consortium via the Proteomics Identification (PRIDE) database [157] (accession PXD030942).

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
