# Peer review of "Multi-Omic Analysis of CIC’s Functional Networks Reveals Novel Interaction Partners and a Potential Role in Mitotic Fidelity"

_cancers, 2023, doi:10.3390/cancers15102805_

Round 1

Reviewer 1 Report

Yuka Takemon and colleagues present an important study identifying the co-essential gene network of the tumor suppressor CIC.  The authors use a combination of directed statistical analyses of well-vetted public perturbation data and genetically engineered human cells in culture to identify and further examine the candidate genes.  Because numerous candidates identified are known interactors, the approach seems sound, and the candidate list likely contains many novel true positives.  Thus, the authors claims that the study greatly expands the knowledge of CIC biology and effects of CIC loss in cancers seems warranted.  The candidate lists provide many new directions for functional studies of CIC.  The study reveals new insight into CIC role in a variety of cellular processes in normal and cancer cells.  It may be more generally useful as an approach as well.

1.      Additional attention to expected false positives (genes that are correlating but not co-essential?) would be useful.  What proportion of the candidates might be false and how can this be estimated? With any candidate gene list, it is very helpful to have a sense of true vs false positives. 

2.      One caveat that should be addressed is whether the HOG cell line truly represents an oligodendroglioma. Based on this report and others, it does not have characteristic genetic alterations of Oligo (CIC, FUBP1, TERT promoter mutations or chr1p/19q codel).  It is not a critical point for the manuscript because many of the CIC interactors are likely shared by multiple cancer cell types, including HEK cells and the NHA.  Along these lines, are any of the known true positive genes cancer type-specific? 

3.      I find the repeated use of possessive (“CIC’s” rather than “of CIC” or similar) awkward, in the title and text. Not sure if it is grammatically correct?

Reviewer 2 Report

“Multi-omic analysis of CIC’s functional networks reveals novel interaction partners and a potential role in mitotic fidelity” by Takemon et al. is a very nice data-heavy manuscript describing the use of multi-omic approach to unveil the complex interactome of the cancer-related gene, CIC. The introduction gave excellent and relevant background information on CIC. The methods section properly listed the materials, bioinformatics tools used and dataset used. I appreciate the use of supplementary methods for listing more details. The results and discussion section are extremely long and a bit wordy in some parts. I really appreciate some of great data in this manuscript, but there are parts that I find really hard to read and absorb efficiently due to overloading. I encourage the author to condense and present some information in the text in a different and efficient format (such as tables). Alternatively, some parts could be moved to the supplementary section to further shorten the main text.

Overall, the manuscript is rich with data and experiments are conducted very well. Therefore, I recommend that the manuscript be published after some improvements in its readability and flow. Please refer below for some specific points.

Results

Line 268-285: The descriptions of the nine co-essential genes are great to have but may be better presented in table format for improved readability. 

Line 290: It would be good to include the ranking of the three selected genes here.

Line 393-402: These lists of genes can be presented in a table or incorporated to Figure 1

Line 456-465: Same as above, presenting this in a table format can help readability

Line 503: It is good to mention the rationale of using PLA as it gives high resolution data and thus a stronger level of evidence for direction interaction than co-IP (which can also pick up indirect interactors through complexes).

Line 587-588: Are these selected phase specific genes themselves affected by CIC status (KO vs WT)? Wouldn’t this potentially confound the cell cycle phase scoring?

Line 625-626: Several of these are in the list of genes used for cycle cell phase scoring right? Shouldn’t they be avoided from being used in the cycle cell scoring? Perhaps if the list of large enough, these effects may not affect sorting of cell cycles.

Line 652-672: These more detailed descriptions of each gene may be better summarized in a table or supplementary table.

Figure 4: How many cells (or % of population) were scored into each cell cycle phases?

Figure 5d-l: The splicing data differences looked quite modest. For the read counts from the different conditions, are these count values normalized (eg. to correct sequencing depth, etc.) before being plotted?

Discussion: While it is true that a big advantage of multi-omics study is the hypothesis generation and broad confirmation, I am curious about some of the limitations in this study. Indeed, multi-functional genes (including CIC) are particularly hard to study. While the multi-omics analyses gain valuable information on the interactome, such complex of intersection of datasets are bound to show potential false positives (and false negatives) as well.

Minor/Editorial:

Line 53: suggest change “inferior” to “reduced”

Line 54: missing comma at the end

Line 620: missing comma after” phases”
